# Vitamin K2 (MK-7) Intercepts Keap-1/Nrf-2/HO-1 Pathway and Hinders Inflammatory/Apoptotic Signaling and Liver Aging in Naturally Aging Rat

**DOI:** 10.3390/antiox11112150

**Published:** 2022-10-30

**Authors:** Mohamed El-Sherbiny, Hoda Atef, Ghada M. Helal, Rasha Hamed Al-Serwi, Hany A. Elkattawy, Gehan Ahmed Shaker, Eman Said, Moaz Abulfaraj, Marzough A. Albalawi, Nehal M. Elsherbiny

**Affiliations:** 1Department of Basic Medical Sciences, College of Medicine, AlMaarefa University, P.O. Box 71666, Riyadh 11597, Saudi Arabia; 2Department of Anatomy, Faculty of Medicine, Mansoura University, Mansoura 35516, Egypt; 3Department of Histology and Cell Biology, Faculty of Medicine, Mansoura University, Mansoura 35516, Egypt; 4Department of Medical Biochemistry, Faculty of Medicine, Mansoura University, Mansoura 35516, Egypt; 5Department of Basic Dental Sciences, College of Dentistry, Princess Nourah Bint Abdulrahman University, P.O. Box 84428, Riyadh 11671, Saudi Arabia; 6Department of Medical Physiology, Faculty of Medicine, Zagazig University, Zagazig 44519, Egypt; 7Department of Medical Physiology, Faculty of Medicine, Mansoura University, Mansoura 35516, Egypt; 8Department of Pharmacology and Toxicology, Faculty of Pharmacy, Mansoura University, Mansoura 35516, Egypt; 9Department of Pharmacology and Toxicology, Faculty of Pharmacy, New Mansoura University, New Mansoura 35511, Egypt; 10Department of Surgery, Faculty of Medicine, King Abdulaziz University, Jeddah 21589, Saudi Arabia; 11Department of Chemistry, Alwajh College, University of Tabuk, Tabuk 71491, Saudi Arabia; 12Department of Pharmaceutical Chemistry, Faculty of Pharmacy, University of Tabuk, Tabuk 71491, Saudi Arabia; 13Department of Biochemistry, Faculty of Pharmacy, Mansoura University, Mansoura 35516, Egypt

**Keywords:** aging, vitamin K2, Nrf-2, COX-2, TIMP, apoptosis

## Abstract

Aging is a naturally occurring physiological process with a deleterious impact on various body organs and humans’ well-being. The aging population is increasing worldwide, which imposes the need for the exploration of nutritional options that can intercept the impact of the aging processed on various body organs. Vitamin K2 (VK2) is a fat-soluble vitamin with emerging evidence on its therapeutic merits. In the current study, natural aging induced a significant liver deterioration with a disrupted Keap-1/Nrf-2/HO-1 axis and increased COX-2, iNOS and TNF-α expression and apoptotic and fibrotic changes. VK2 administration, on the other hand, improved the biochemical indices of liver function (total protein, albumin, ALT and AST); the suppressed hepatic expression of Keap-1 and increased the hepatic expression of Nrf-2 with a parallel increase in the hepatic activity of HO-1. Subsequently, the liver content and hepatic expression of TNF-α, COX-2 and iNOS were significantly retracted. In context, the liver content and hepatic expression of the fibrotic biomarkers TGFβ and TIMP significantly retracted as well. Moreover, the TUNEL assay confirmed the retraction of liver apoptotic changes. Of notice, electron transmission microscope examination confirmed the preservation of mitochondrial functions and preservation of the ultra-microscopical structures. In conclusion, the VK2-mediated interception of aging-induced Keap-1/Nrf-2/HO-1 signaling suppressed the hepatic contents of inflammatory and fibrotic biomarkers, as well as apoptotic changes with preservation of the hepatic architectural and functional status. VK2 can be presumed to be an effective nutritional supplement to the aging population to spare the liver, amongst other body organs, against aging-induced deleterious injury.

## 1. Introduction

Aging or senescence is a natural and multifactorial process characterized by progressive decline in the physiological functions, with subsequent adverse consequences on various body organs [1]. Indeed, the age of the global population has been constantly increasing, imposing age-related health burdens [2]. Therefore, aging societies represent a considerable problem for healthcare systems. The aging process is associated with a gradual dysregulation of genomic, cellular, tissue and whole-organism homeostasis, leading to a compromised ability to respond to stress, deteriorated function and increased morbidity and mortality risks. Moreover, these changes lead to serious pathologies, including cardiovascular disorders, diabetes, neurodegenerative diseases and cancer [3].

Aging signs vary among body organs, depending on their difference in anatomy and functions. The liver plays a major role in metabolism and metabolic processes. Cellular senescence affects various types of liver cells, including hepatocytes, Küpffer cells, the hepatic stellate and the liver sinusoidal endothelial cells [4]. The genome and epigenomic alterations, mitochondrial dysfunction, altered intercellular communication and dysregulated nutrient sensing pathways have been reported to contribute to hepatic cellular senescence and, ultimately, the impairment of hepatic functions [5]. In this context, various cellular signaling pathways have been reported to be dysregulated in aged liver, including oxidative stress and inflammation [6,7]. The Nuclear factor erythroid 2–related factor 2 (Nrf-2)/Kelch-like ECH-associated protein 1 (Keap-1)/heme oxygenase 1 (HO-1) axis is a known conserved intracellular defense pathway to combat oxidative stress [8,9]. A reduced Nrf-2/Keap-1/HO-1 axis reflects disrupted cellular redox homeostasis [10]. Redox imbalance is accompanied by increased inducible nitric oxide synthase (iNOS) with a subsequent increase in nitric oxide and peroxynitrite free radicals [11]. Other changes, including increased cyclooxygenase-2 (COX-2) expression and elevated inflammatory cytokines production, have also been reported in aged liver [12]. All these events contribute to increased extracellular matrix proteins accumulation accompanied by deteriorated liver functions. The tissue inhibitor of metalloproteinase (TIMP-1) has been correlated to age-dependent liver fibrosclerosis [13]. Additionally, TGF-β is a reliable marker of liver fibrosis [14]. Tumor growth factor-β (TGF-β) modulates hepatic cells plasticity, induces hepatocytes epithelial–mesenchymal transition and activates hepatic stellate cells, leading ultimately to hepatic fibrosis [15]. These molecular events could be targeted to attenuate age-induced structural and functional alterations of the liver.

Vitamin K (VK) is a fat-soluble vitamin crucially involved in blood homeostasis. It acts as cofactor for the γ-carboxylase enzyme. It exists in two forms, VK1 or phylloquinone and VK2/MK-n or menaquinones [16]. VK1 is found in plant sources, including green leafy vegetables and vegetable oils, while VK2 is abundant in animal-based and fermented products [17]. MK-7 can be found in adequate amounts in fermented milk-based and soybean products. Moreover, animal products such as meat, eggs and fish are rich in MK-4 [18]. VK has also been reported to modulate bone homeostasis [19], suppress inflammation by inhibiting the nuclear factor kappa B (NF-κB) signaling pathway [20], protect against oxidative stress by blocking reactive oxygen species generation [21] and phosphorylate protein kinase A, favoring anticancer activity [22]. 

In the liver, VK has been suggested to be effective against hepatic failure and cholestatic liver disease. VK1 decreased the mortality risk in patients with chronic liver failure [23]. Moreover, VK deficiency has been reported in patients with hepatocellular carcinoma (HCC), and VK2 analogs suppressed HCC development in patients with cirrhosis [24], as well as HCC recurrence in patients with hepatitis [25]. 

Interestingly, many studies supported a protective efficacy for VK in age-related conditions [26]. In this context, the protective effect of MK-7 against age-induced structural changes and cognitive deficit in naturally aged rats has been referred to [27]. However, the protective effect of VK against structural and functional changes in aged liver has not been investigated. The incidence of age-related diseases is currently increasing because of the rising human life expectancy. Modifying lifestyles and including certain nutritional options can be presumed to attenuate aging-induced pathological changes. Hence, nutritional supplements capable of enhancing the quality of life, promote healthy aging and modulate the prognoses of age-related diseases are of considerable importance [28]. Considering the reported efficacy of VK in age-related diseases, the present study aimed to evaluate the potential protective effect of VK (MK-7) against liver injury in naturally aging rats. Moreover, the possible underlying mechanisms have been investigated.

## 2. Materials and Methods

### 2.1. Animals and Experimental Design

Twenty adult male Sprague–Dawley rats aged three months were provided by “the Holding Company for Biological Products and Vaccines”, “VACSERA”, Egypt. The animals were first acclimatized for one week under standard laboratory conditions. Then, the rats were randomly assigned into two groups (*n* = 10): Aged untreated rats: rats received sunflower oil (vehicle) for 17 months, and Aged-treated rats: rats received VK2 (MK-7) at a dose of 30 mg/kg 5 days per week for 17 months [27]. An additional group of adult rats (three months age, *n* = 10) was used as the adult control group.

At the end of the experiment, blood samples were withdrawn and centrifuged, and sera were further used for biochemical analyses of liver function indices. The abdominal cavity was opened, and the livers were dissected, washed in ice-cold phosphate-buffered saline (PBS), blotted dry using filter papers and divided into four parts; the first part was homogenized and used for protein analysis by Enzyme-Linked Immunosorbent Assay (ELISA), the second part was flash-frozen and stored at −80 °C to be used for real-time polymerase chain reaction (RT-PCR), the third part was fixed in buffered formalin for histological and immunohistochemical assessments and the fourth part was fixed in buffered glutaraldehyde for ultrastructure examination by a transmission electron microscope. 

### 2.2. Biochemical Analysis of Liver Function Tests

Serum alanine aminotransaminase (ALT, ELITech Clinical Systems, Zone Industrielle, Sées, France), aspartate aminotransaminase (AST, ELITech Clinical Systems, Zone Industrielle, Sées, France), total protein (Biodiagnostic, Dokki, Giza, Egypt, TP 20 20) and albumin (Biodiagnostic, Dokki, Giza, Egypt, AB 10 10) were quantified using a commercially available kit following the manufacturers’ instructions.

### 2.3. Liver HO-1 and TIMP Assay by ELISA

Hepatic tissue homogenates (10% *w/v*) were prepared in chilled 1.15% KCl at pH 7.4 using a handheld homogenizer (Omni international, Kennesaw, GA, USA). The homogenates were further analyzed for HO-1 (BioVision, Milpitas, CA, USA, # E4525-100) and TIMP-1 (BioVision, Milpitas, CA, USA, # E4844-100) following the manufacturers’ protocols.

### 2.4. Real-Time RT-PCR Assay of Liver Nrf-2 and Keap-1

Direct-zol RNA Miniprep Plus (Cat# R2072, ZYMO RESEARCH CORP., Irvine, CA, USA) was used to extract RNA from homogenized hepatic tissues from all different groups according to the manufacturers’ protocols. The extracted RNA was assessed for quantity and quality using a Beckman dual spectrophotometer (USA). RNA was then used to prepare complementary DNA (cDNA) using the SuperScript IV One-Step RT-PCR kit (Cat# 12594100, Thermo Fisher Scientific, Waltham, MA USA). Real-time PCR was then performed using 96-well plate StepOne instrument (Applied Biosystem, Waltham, MA, USA). The thermal profile used was as follows: reverse transcription at 45 °C for 10 min, RT inactivation at 98 °C for 2 min, 40 cycles initial denaturation at 98 °C for 10 s and an amplification step at 55 °C for 10 s and at 72 °C for 30 s. The data were presented as the cycle threshold (Ct) for target and housekeeping gene. To normalize the variations in the target genes expression, *Nrf2* and *Keap-1,* the ΔΔCt method was performed referring to the mean CT expression values of housekeeping gene *GAPDH*. The relative quantitation (RQ) of each target gene was calculated using the 2^−∆∆Ct^ method. Sequence of the primers for the *Nrf2* gene was: forward 5′-TCCCAAACAAGATGCCTTGT-3′ and reverse 5′-AGAGGCCACACTGACAGAGA-3′ (GeneBank accession number is NM_031789.2); the *Keap-1* gene was: forward 5′-CGCCCTCGCAGGATGGTAAC-3′ and reverse 5′-AGGCATCCACCCCAAGGTCA-3′ (GeneBank accession number is NM_057152.2) and the *GAPDH* housekeeping gene was: forward 5′-CCTCGTCTCATAGACAAGATGGT-3′ and reverse 5′-GGGTAGAGTCATACTGGAACATG-3′ (GeneBank accession number is NM_001394060.2).

### 2.5. Histological Study

Fixed liver specimens were dehydrated using alcohol solutions of ascending grades. Following dehydration, the tissue specimens were embedded in paraffin and then cut into 5-μm sections using a rotary microtome. Sections were then deparaffinized prior to staining with hematoxylin and eosin (H&E). The slides were further inspected for any structural changes, and images were captured using an Olympus microscope with built-in camera (Olympus Optical Co., Ltd., Tokyo, Japan).

### 2.6. Immunohistochemical (IHC) Analysis of Tumor Necrosis Factor-α (TNF-α), COX-2, iNOS and TGF-β1

Additional hepatic deparaffinized sections were processed for IHC. Sections were incubated for 30 min with 0.3% H_2_O_2_ to abolish endogenous peroxidase activity. Then, the slides were incubated at 4 °C overnight with primary antibodies purchased from Wuhan Servicebio Technology, Wuhan, China, for TNF-α, (GB11188, 1:500), COX-2, (GB11077-1, 1:500), iNOS, (GB11119, 1:500) and TGF-β (FNab08638, 1:500). Thereafter, the slides were washed and incubated with appropriate labeled goat anti-rabbit secondary antibodies (G1213, 1:200). 3,3′-diaminobenzidine was used for visualization. ImageJ software (NIH, MD, USA) was used to evaluate the mean % area of immunostaining. An Olympus^®^ digital camera fixed on an Olympus light microscope (Tokyo, Japan) was used to capture images at 400× magnification.

### 2.7. Transmission Electron Microscope

Glutaraldehyde fixed hepatic specimens were treated with 1.0% osmium tetroxide for one hour. The specimens were then washed and treated with alcohol solutions of different grades for dehydration. Then, the dehydrated specimens were embedded in resin composed of propylene and epoxy in a 1:1 ratio, followed by epoxy resin. Ultra-thin sections were then cut and stained with lead citrate and uranyl acetate. Images were captured by transmission electron microscope (JEOL 2100, Tokyo, Japan).

### 2.8. DNA Fragmentation

DNA fragmentation is a hallmark of apoptosis. The ApoBrdU DNA Fragmentation Assay Kit (Catalog#: K401) was used for the assessment of apoptosis in hepatic tissue, according to the manufacturers’ protocol.

### 2.9. Statistical Analysis

The results were presented as the mean ± SE. One-way analysis of variance (ANOVA) was used for statistical comparisons, followed by Tukey’s post hoc test. Graphical presentation of the results and statistical analysis were performed using GraphPad Prism 8 (GraphPad Software, San Diego, CA, USA).

## 3. Results

### 3.1. Biochemical Analysis of Liver Functions

Compared to the adult control group, the aged control rats demonstrated a significant deterioration in liver function biomarkers. The serum ALT and AST were significantly higher by approximately 1.52- and 1.8-fold, respectively, while the serum total protein and albumin significantly decreased by 0.12- and 0.19-fold, respectively, compared to the adult control group. VK2 administration reversed this situation, as confirmed by the significant decrease in serum ALT and AST by 0.25- and 0.27-fold, respectively, with the parallel significant restoration of the serum total protein and albumin by 1.1- and 1.13-fold, respectively, reaching values close to the adult control (Figure 1). 

### 3.2. Effect of VK2 on Aging-Induced Hepatic Cellular Structural and Ultrastructural Changes

H&E-stained sections from the adult control revealed preserved hepatic architecture. On the other hand, hepatic cellular injury was evident in the untreated aged controls characterized by the irregular arrangement of the hepatocytes with vacuolated cytoplasm. Apoptotic changes with dark small nuclei were observed in some cells, while others showed degenerative nuclear changes. Areas of piecemeal necrosis and increased portal inflammatory infiltrate were seen. Neocholagioles and areas of focal lytic necrosis were also detected. Moreover, shrunken central veins with blood congestion and irregular blood sinusoids were also detected. However, sections from VK2-treated rats demonstrated almost normal hepatic architecture with rounded hepatocyte nuclei, intact endothelial lining of the central vein and preserved structure of the portal tract (Figure 2).

Regarding the ultrastructure examination, TEM confirmed irregular nuclei, rarefied cytoplasm with multiple vacuoles and lipid droplets, hepatocytes’ nuclear degenerative changes and disrupted or lost hepatocytes’ microvilli in the aged control rats. These changes were ameliorated in VK2-treated rats, where hepatic sections showed hepatocyte with regular nuclei, restored microvilli and elongated normal mitochondria (Figure 3).

### 3.3. Effect of VK2 on Hepatic Keap-1/Nrf-2/HO-1 Axis in Aged Rats

The mRNA level of Nrf-2 was significantly downregulated by 0.76-fold, and Keap-1 was significantly increased by 1.47-fold in hepatic tissue of the aged untreated controls compared to adult controls. This was accompanied by a marked decrease in the HO-1 hepatic protein levels by 0.68-fold. On the other hand, VK2 administration modulated the age-induced suppression of the Keap-1/Nrf-2/HO-1 axis, as indicated by the significant restoration of the Nrf-2 mRNA level and HO-1 protein by 3.3- and 3.13-fold, respectively, along with the significant decrease in Keap-1 mRNA by 0.32-fold in the hepatic tissue of treated aged rats (Figure 4). 

### 3.4. Effect of VK2 on Hepatic COX-2, iNOS and TNF-α

For the assessment of inflammation among different experimental groups, COX-2, iNOS and proinflammatory cytokine TNF-α immunostaining were evaluated. As shown in Figure 5, aged untreated control rats demonstrated a significant increase in hepatic COX-2, iNOS and TNF-α immunostaining compared to the adult control. In contrast, VK2 administration to aged animals attenuated hepatic inflammation where hepatic sections from aged-treated rats demonstrated a marked downregulation in COX-2, iNOS and TNF-α immunostaining in hepatic tissue when compared to the aged untreated group.

### 3.5. Effect of VK2 on Hepatic TIMP-1 and TGF-β in Aged Rats

The aged untreated control rats in the present study demonstrated a significant increase in hepatic TIMP-1 concentration with a significant upregulation of hepatic TGF-β immunostaining compared to the adult controls. On the other hand, VK2-treated aged rats showed a significant downregulation in both hepatic TIMP-1 concentration and TGF-β immunostaining compared to the aged untreated control (Figure 6).

### 3.6. Effect of VK2 on Hepatic Apoptosis in Aged Rats

Figure 7 displays the results of the TUNEL assay of apoptosis. Aged untreated control rats demonstrated a significant increase in the % of the TUNEL-positive cells in hepatic sections, indicating increased apoptotic cells in comparison with the adult control. In contrast, VK2-treated aged rats showed a significant decrease in the % of the TUNEL-positive cells in hepatic sections compared to the aged untreated control.

## 4. Discussion

The total number of people aged 65 or more is about 8% of the total global population, and it is expected to further increase to about 19% by 2030 [29]. This is attributed to remarkable developments in both health and medical services; however, aging itself is accompanied by disrupted homeostasis, with an increased risk for chronic diseases [5]. In liver, aging has been reported to be associated with gradual alteration of the hepatic structure and function [30]. It also increases the risks for various liver diseases and their mortality rates [31,32]. Additionally, aged liver possesses a decreased regenerative ability with delayed restoration of the liver function [33]. Similarly, the results of the current study demonstrated an alteration in the liver architecture, apoptotic changes, necrosis and hepatic inflammatory changes in aged animals. 

Aged liver demonstrates a decreased blood flow and cellular size. This is accompanied by oxidative stress-induced protein denaturation and the subsequent accumulation of undegradable protein aggregates. These events induce the further generation of reactive oxygen species (ROS) with subsequent reduced cell survivability [34]. Indeed, aging disrupts the oxidants/antioxidants balance in normal cells, and the excessive generation of ROS products activates the damage of macromolecules, leading ultimately to DNA mutations [35]. Likewise, aged liver demonstrates oxidative stress that, in turn, induces inflammatory signaling, accelerated cellular senescence and, eventually, organ dysfunction [36]. The current study revealed the confirmatory results, as liver aging was associated with an escalation in liver function biomarkers ALT and AST, with retraction in its synthetic functions, as confirmed by the significant retraction in the serum total protein and albumin. Nevertheless, impaired liver oxidative/antioxidative status was evident, as observed in the significant downregulation in hepatic Nrf-2 expression and HO-1 activity with the parallel increase in keap-1. Moreover, the hepatic expression of COX-2, TNF-α and iNOS were markedly increased, thus confirming the intricate interaction between oxidative/inflammatory signaling in the progression of aging-induced liver injury. 

During the aging process, mitochondrial dysfunction results in a decrease in energy production and increase in the oxidation of proteins, DNA and phospholipids [37]. In agreement, the results of the current study also confirmed the incidence of mitochondrial and hepatocytes nuclear degenerative changes, as well as DNA fragmentation, as evident by the electron microscopic examination and the TUNEL assay of liver apoptosis. Apoptosis is significantly altered during the aging process [38]. Nevertheless, the activation of hepatic stellate cells and hepatocyte apoptosis participates in the progression of liver fibrosis. Indeed, excessive hepatocyte apoptosis induces ECM accumulation with the progressive development of hepatic fibrosis [39]. 

Aging has been suggested to increase the susceptibility of liver fibrosis [40]. The fibrogenic response to chronic CCl_4_ injury was reported to be significantly higher in the livers of older rats compared to the younger ones [41]. In line with this, the results of the current study confirmed increased markers of fibrosis in the livers of the aging rats, as evident by the increased hepatic TGF-β1 and TIMP-1 expression.

With the increased number of the globally aging population, as previously referred to, comes the increased need to explore novel approaches that will decrease or attenuate the consequences of the aging processes on the body organs and, especially, the liver, given its key functions in the excretory, metabolic and synthetic functions. 

VK2 is a common fat-soluble vitamin involved in a variety of metabolic processes [42]. In the current study, VK2 administration to the naturally aging rats hindered the deleterious effects of the aging liver and the liver functions, as confirmed by the observed retraction in the liver biomarkers, ALT, AST and the restoration of the serum total protein and albumin concentrations. Moreover, the observed improvement in the histopathological image and the restoration of normal hepatic architecture of the aging rats on VK2 confirms the positive impact of VK2 in hindering the aging process. The significant improvement of the hepatic functions and architecture with VK2 administration was associated with a coordinated modulation of oxidative/inflammatory and apoptotic signaling in the liver.

Emerging evidence reported Nrf-2 signaling and VK to play a crucial role in counteracting oxidative stress, DNA damage, senescence and inflammation [43]. In the current study, VK2 administration induced a significant restoration in the hepatic expression of Nrf-2 and HO-1 with suppression of the hepatic expression of Keap-1. Keap1 is a well-known inhibitor protein of Nrf2. Nrf2 is a key regulator of the cellular response to oxidative stress [44,45]. Under normal physiological conditions, Keap1 binds to Nrf2 to form a complex in a low activity state. In contrast, various oxidative and/or pathological insults induces the modification of Keap1 cysteine residue, triggering Nrf2 phosphorylation and translocation to the nucleus, where it mediates the transcription process to control the downstream gene expression of a series of antioxidant enzymes of HO-1 [46]. These events help in quenching ROS; removing other cellular harmful substances and activating the antioxidative defense mechanism, antiapoptotic, anti-inflammatory and other cell protection mechanisms [47].

Herein, it can be presumed that VK suppressed Keap-1, which, in turn, enhanced Nrf-2 translocation to the nucleus and the enhanced HO-1 encoding and activation with the subsequent enhancement of antioxidative signaling in the liver. The modulation of the hepatic expression of hepatic Keap-1/Nrf-2/HO-1axis was associated with a significant retraction in the hepatic expression of inflammatory biomarkers COX-2, TNF-α and iNOS and fibrotic biomarkers TGF-β and TIMP, retraction in hepatic apoptosis and significant improvement in the hepatic architectural image with the ultimate restoration of normal liver functions. 

Indeed, VK2 administration has been reported to restore brain Nrf-2 contents in naturally aging rats and, hence, improve aging-induced structural and cognitive functions [23]. Thus, it can be presumed that VK2-mediated interception of aging-induced Keap-1/Nrf-2/HO-1 signaling conferred significant anti-inflammatory, antifibrotic and antiapoptotic impacts, which, in turn, hindered the deleterious impacts of the naturally occurring aging process on the liver.

VK was reported to exert an important role in cellular senescence, which was attributed to its antioxidant and anti-inflammatory effects. VK was reported to be a strong scavenger of free radicals, demonstrating a 10–100-times more potent antioxidant effect when compared to other antioxidants, including vitamin E [48]. Interestingly, NAD(P)H Quinone Dehydrogenase 1 (NQO1), a classic antioxidant protein known to be encoded by a Nrf2 target gene, catalyzes VK reduction, serving as a link between Nrf2 signaling and the VK cycle [49]. Noteworthy, VK2 rescued the mitochondrial dysfunction in *Drosophila* via the restoration of ATP production [50].

Nevertheless, the VK2 activation of Gas-6 and Protein S provided protection against neuronal ischemic/hypoxic injury via antiapoptotic and anti-inflammatory effects [51]. In context, VK2 successfully restored the oxidative/antioxidative balance with a parallel suppression of inflammatory genetic expression in the brain of naturally aging rats [27], which gives further credence to the observations of the current study. Moreover, VK2 suppressed TNF-α in human lymphocytes, fibroblasts and astrocytes, where inflammation is induced by various agents [52].

VK2 administration to the aging rats in the current study hindered the upregulation of fibrotic markers in liver tissue. It can be presumed that the VK2-mediated suppression of oxidative, inflammatory and apoptotic signaling participate in its effects on markers of liver fibrosis.

## 5. Conclusions

VK2-mediated interception of aging-induced Keap-1/Nrf-2/HO-1 signaling suppressed the hepatic contents of inflammatory biomarkers COX-2, TNF-α and iNOS and fibrotic biomarkers TGF-β and TIMP and apoptosis with the preservation of hepatic architectural and functional status. Thus, VK2 can be presumed to be an effective supplement to the aging population to prevent aging-induced deleterious effects on the liver amongst other body organs. 

## Figures and Tables

**Figure 1 antioxidants-11-02150-f001:**
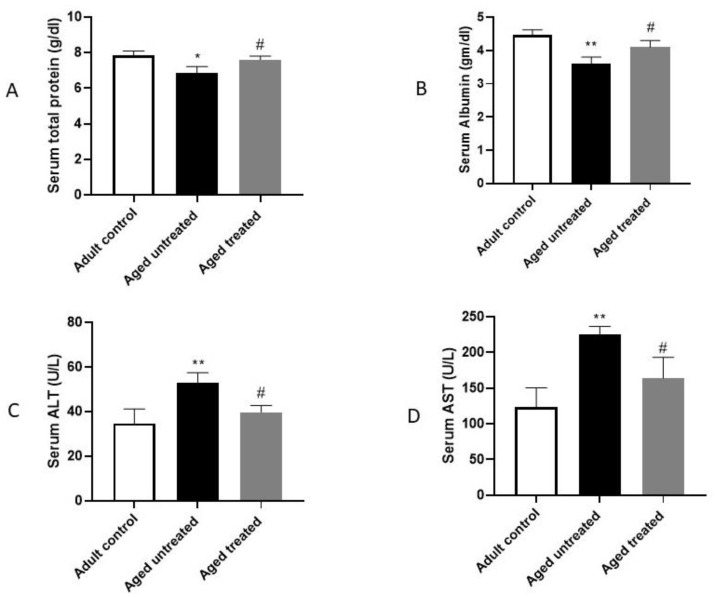
Effect of VK2 administration on liver function tests. (**A**) Serum total protein, (**B**) Serum Albumin, (**C**) Serum alanine aminotransferase (ALT) and (**D**) Serum aspartate aminotransferase (AST). Values are expressed as the mean ± SE, *n* = 6. * *p* < 0.05 and ** *p* < 0.01 compared to the Adult control group. ^#^ *p* < 0.05 compared to the Aged untreated group.

**Figure 2 antioxidants-11-02150-f002:**
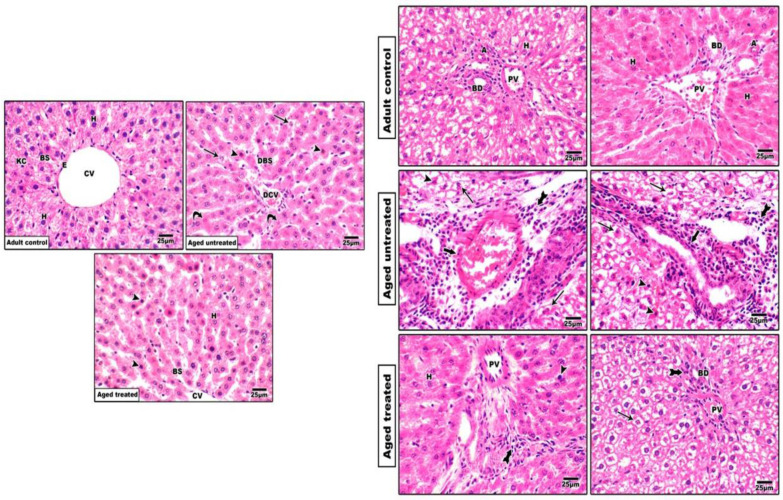
Left panel showing central vein: Adult control group showing hepatocyte cords radiating from the central vein with acidophilic cytoplasm and rounded nuclei. Aged untreated group showing an irregular arrangement of hepatocytes with vacuolated cytoplasm. Some cells show apoptotic changes with dark small nuclei. Aged-treated group showing hepatocyte-arranged cords with fewer apoptotic cells. H: hepatocytes, Head arrows: apoptotic bodies, CV: Central vein, Arrows: degenerated hepatocytes, DBS: dilated blood sinusoids and DCV: dilated central vein. Right panel showing portal vein: Adult control group showing normal hepatocytes with acidophilic cytoplasm and rounded nuclei around the portal tract composed of the portal vein, hepatic artery and bile duct. Aged untreated group showing an irregular arrangement of hepatocytes with marked vacuolization and hydropic degeneration. Areas of piecemeal necrosis and increased portal inflammatory infiltrate are seen. Some cells show apoptotic changes with dark small nuclei. Neocholagioles and areas of focal lytic necrosis are present. Aged-treated aged group showing hepatocytes and portal tract with fewer apoptotic cells and inflammatory infiltrate. Reduced areas of lytic necrosis and a portal tract-preserved structure are also observed. H: normal hepatocytes, Arrows: degenerated hepatocytes, Head arrows: apoptotic cells, PV: portal vein, A: hepatic artery, BD: bile duct, bold arrow: inflammatory infiltrate and tailed arrows: focal lytic necrotic areas.

**Figure 3 antioxidants-11-02150-f003:**
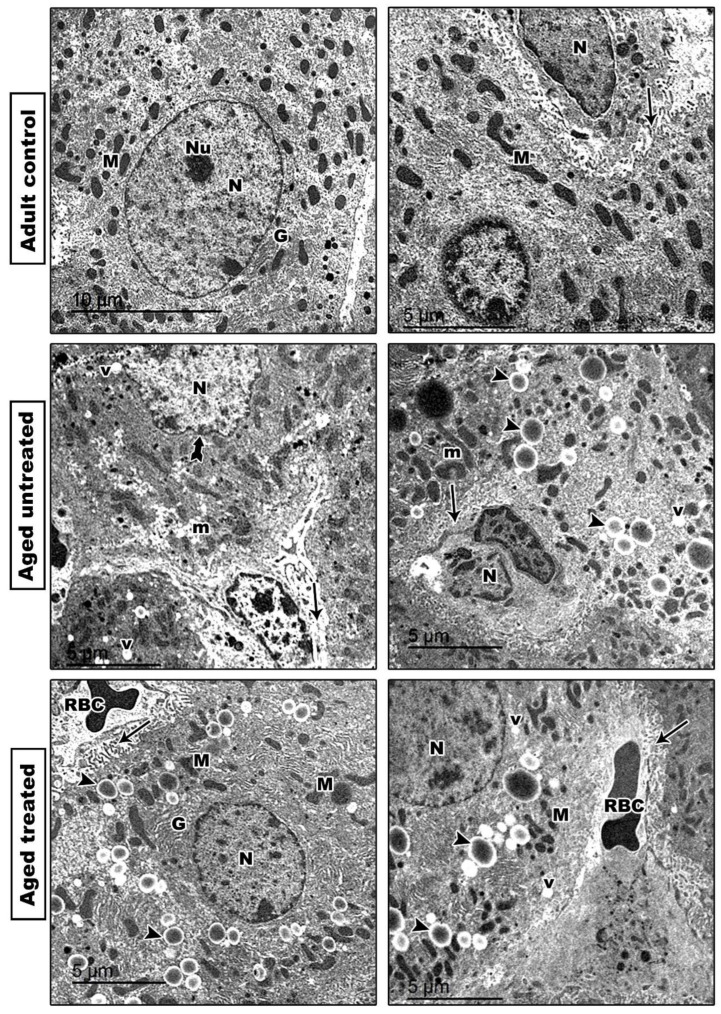
Transmission electron photomicrograph of hepatocytes showing: A1 and A2: Adult control hepatocytes with euchromatic nucleus, prominent Golgi apparatus and numerous elongated mitochondria. Abundant microvilli of hepatocytes are noted in the perisinusoidal space of Kupffer cells are also shown with their irregular dark nuclei. B1 and B2: Aged untreated hepatocytes showing irregular nuclei and rarefied cytoplasm with multiple vacuoles and lipid droplets. Hepatocytes nuclear degenerative changes are noted. Microvilli of hepatocytes are disrupted or lost. C1 and C2: VK2-treated aged group showing hepatocyte with regular nuclei, restored microvilli of hepatocytes and elongated normal mitochondria. N: nucleus, Tailed arrow: irregular nucleus, M: normal mitochondria, m: degenerated mitochondria, V: Cytoplasmic vacuoles, Arrows: microvilli, G: Golgi complex and Head arrows: lipid droplets.

**Figure 4 antioxidants-11-02150-f004:**
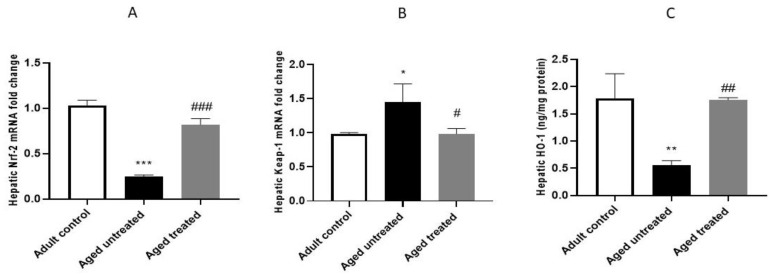
Effect of VK2 administration on the Keap-1/Nrf-2/HO-1 axis in hepatic tissues. (**A**) Hepatic Nrf-2, (**B**) hepatic Keap-1 mRNA fold change, as detected by RT-PCR and (**C**) hepatic HO-1 concentration (ng/mg protein) as detected by ELISA. Values are expressed as the mean ± SE, *n* = 6. * *p* < 0.05, ** *p* < 0.01 and *** *p* < 0.001 compared to the Adult control group. ^#^ *p* < 0.05, ^##^ *p* < 0.01 and ^###^ *p* < 0.001 compared to the Aged untreated group.

**Figure 5 antioxidants-11-02150-f005:**
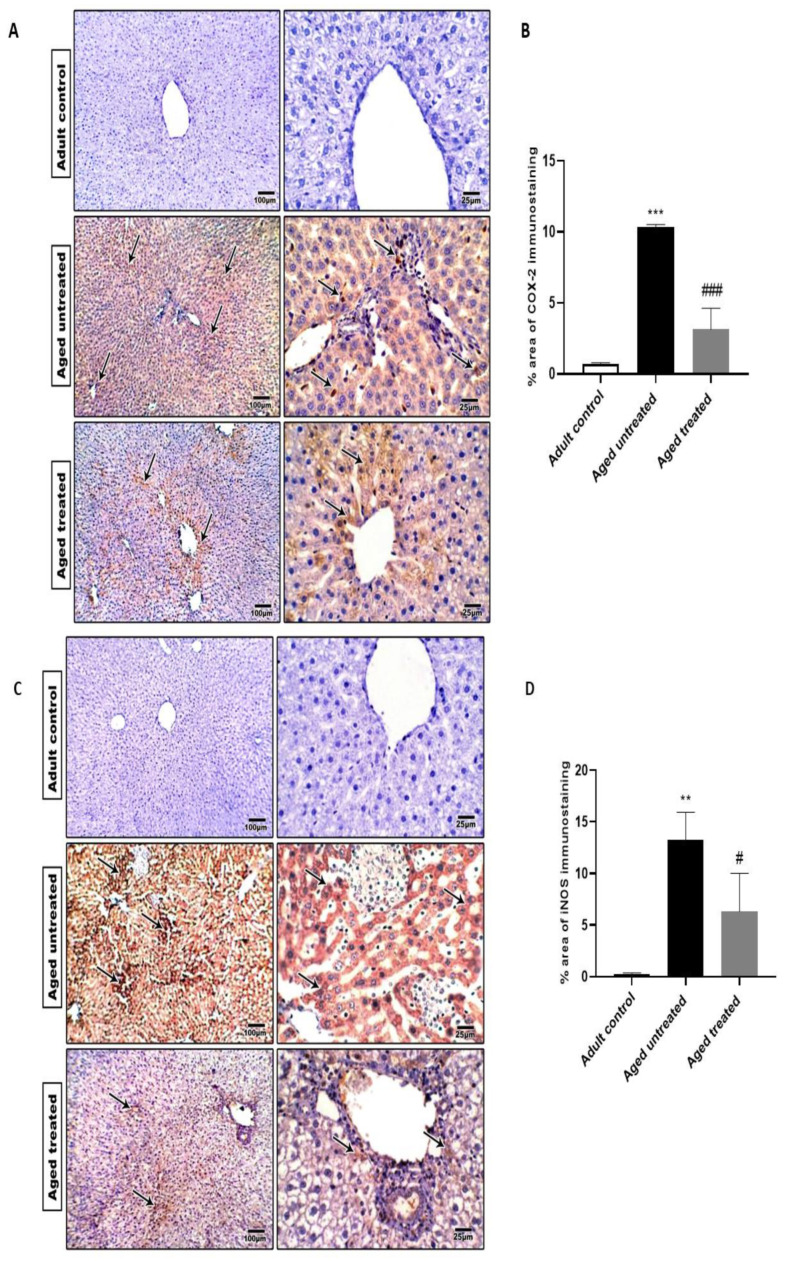
Effect of VK2 administration on hepatic inflammation. (**A**) Hepatic COX-2 immunostaining. (**B**) Quantitative assessment of the % area of hepatic COX-2 immunostaining, (**C**) hepatic iNOS immunostaining, (**D**) quantitative assessment of the % area of hepatic iNOS immunostaining, (**E**) hepatic TNF-α immunostaining and (**F**) quantitative assessment of the % area of hepatic TNF-α immunostaining. Values are expressed as the mean ± SE, *n* = 6. ** *p* < 0.01 and *** *p* < 0.001 compared to the Adult control group. ^#^ *p* < 0.05, ^##^ *p* < 0.01 and ^###^ *p* < 0.001 compared to the Aged untreated group.

**Figure 6 antioxidants-11-02150-f006:**
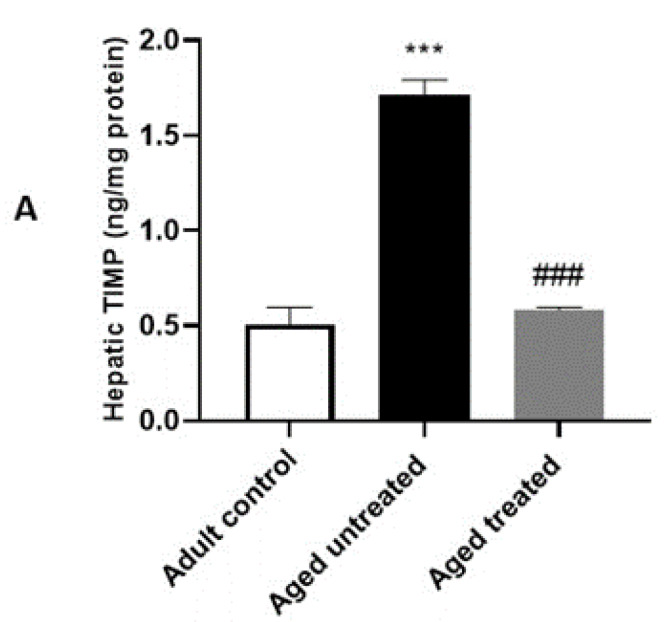
Effect of VK2 administration on markers of fibrosis in hepatic tissue. (**A**) Hepatic tissue inhibitor of metalloproteinase (TIMP) concentration (ng/mg protein), (**B**) hepatic TGF-β immunostaining and (**C**) quantitative assessment of the % area of hepatic TGF-β immunostaining. Values are expressed as the mean ± SE, *n* = 6. *** *p* < 0.001 compared to the Adult control group. ^##^ *p* < 0.01 and ^###^ *p* < 0.001 compared to the Aged untreated group.

**Figure 7 antioxidants-11-02150-f007:**
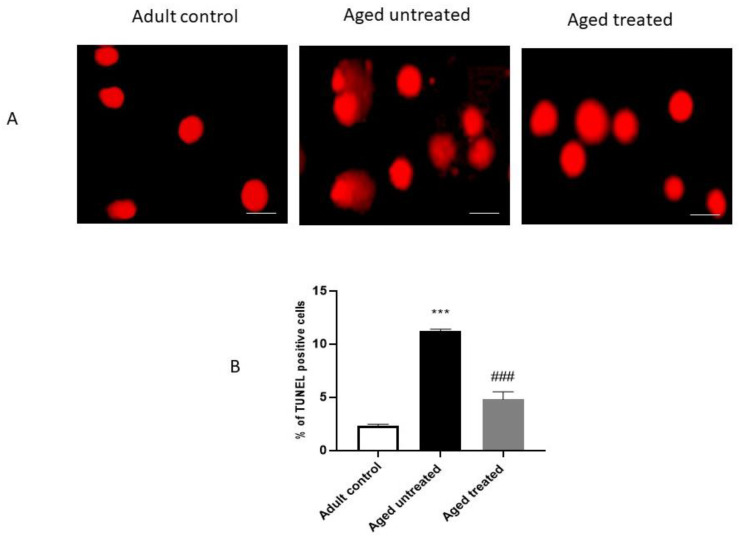
Effect of VK2 administration on hepatic apoptosis. (**A**) TUNEL staining in hepatic sections from different experimental groups. Scale bar (50 nm). (**B**) Percentage of TUNEL-positive cells. Values are expressed as the mean ± SE, *n* = 4–6. *** *p* < 0.001 compared to the Adult control group. ^###^ *p* < 0.001 compared to the Aged untreated group.

## Data Availability

Data are available upon request.

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
