# Peer review of "Vitamin K2 (MK-7) Intercepts Keap-1/Nrf-2/HO-1 Pathway and Hinders Inflammatory/Apoptotic Signaling and Liver Aging in Naturally Aging Rat"

_antioxidants, 2022, doi:10.3390/antiox11112150_

Round 1

Reviewer 1 Report

In this study entitled “Vitamin K (MK-7) Intercepts Keap-1/Nrf-2/Ho-1 pathway and Hinders Inflammatory/Apoptotic signalling and Liver Aging in 3 Naturally Aging Rat” was investigated the eventual protective effect of this vitamin against liver injury in naturally aging rats. The authors used different experimental approaches as well as histological study, TEM, immunohistochemistry to study morphology of the liver in the different conditions and biochemical analysis, TUNEL-assay and real-time RT-PCR to study biomarkers of inflammation, fibrosis and apoptosis. They demonstrated that VK2 improved liver functions, suppressed hepatic expression of Keap-1 and increased hepatic expression of Nrf-2 with paralleled increase in hepatic activity of HO-1. Subsequently, liver content and hepatic expression of the inflammatory biomarkers; TNF-α, COX-2 and iNOS significantly retracted. In context, the liver content and hepatic expression of the fibrotic biomarkers TGFβ and TIMP significantly retracted as well. Moreover, TUNEL assay confirmed the retraction of liver apoptotic changes. They concluded that VK2 can be presumed to  be an effective nutritional supplement to aging population to spare the liver amongst other body organs against aging-induced deleterious injury. 

The topic of this study is interesting, however, it can be accepted after major revision.

I have itemized my major concerns in the following paragraphs.

Abstract

Line 41: eliminate “:” and insert “,”

Introduction :

Improve the aim of the study explaining also the choice of the various markers of inflammation, fibrosis, even with bibliographical references .... avoiding doing it in the result section.

Materials and methods

Line 119: What is “MMP”? Delete if it is a mistake; 

Line 120 delete “:” and check throughout the  manuscript 

Line 121: Please add information regarding the homogenization buffer 

Line 123 : Add “real-time” before RT-PCR because In this paragraph , the authors describe the real-time RT-PCR.

Line 158-159: add information regarding the secondary antibodies

Results

Line 188: “figure1” in bold 

Line 238-240: Delete this sentence “ Nrf-2/Keap-1/HO-1 axis is a conserved intracellular defense pathway to combat oxidative stress” from this paragraph and transfer it in the introduction section.

Line 240-246: This paragraph is difficult to understand as a graph (HO-1 protein level) is missing from figure 4. In addition, the authors must choose whether to use the percentage or the increment and decrement fold for the mRNA levels, otherwise it creates confusion. 

If the authors add the graph, they must also insert the ELISA assay description in the methods section.

Line 254-256 and Line 271-272: Delete and Transfer these sentences in the introduction. 

Figure 2: There is a problem in this figure. What are the images in the left side of this figure ??? in the figure legends the authors describe only the images in the right side (A1-C2). If it is a mistake, delete these images or alternatively describe these in the figure legend and in the paragraph. 

Figure4: Where is C? Miss the graph of ELISA as describe in the figure legend

Figure 5: This figure in not clear in particular for immunohistochemistry where the different images were not specify in the figure legend or in the relative paragraph of the results section (i.e. A1 and A2 of the I section are adult control ???...)

Figure 6: Miss “I” and “II” in the figure 

Discussion

The discussion section  is too long especially in the first part concerning the effects of aging. 

Line: 296-364 . Reduce the discussion summarizing some parts

Line 345 : “TUNEL” and not “TUNNEL”

Line 417: “(13)” is not correct for reference style. 

Author Response

First, we would like to thank the reviewer for dedicating the time and the effort to review our work.  The reviewer’s suggestions were very valuable and significant for the revision and improvement of our article quality and provided very important guidance to our research. The points that were raised by the reviewer are addressed below

In this study entitled “Vitamin K (MK-7) Intercepts Keap-1/Nrf-2/Ho-1 pathway and Hinders Inflammatory/Apoptotic signalling and Liver Aging in 3 Naturally Aging Rat” was investigated the eventual protective effect of this vitamin against liver injury in naturally aging rats. The authors used different experimental approaches as well as histological study, TEM, immunohistochemistry to study morphology of the liver in the different conditions and biochemical analysis, TUNEL-assay and real-time RT-PCR to study biomarkers of inflammation, fibrosis and apoptosis. They demonstrated that VK2 improved liver functions, suppressed hepatic expression of Keap-1 and increased hepatic expression of Nrf-2 with paralleled increase in hepatic activity of HO-1. Subsequently, liver content and hepatic expression of the inflammatory biomarkers; TNF-α, COX-2 and iNOS significantly retracted. In context, the liver content and hepatic expression of the fibrotic biomarkers TGFβ and TIMP significantly retracted as well. Moreover, TUNEL assay confirmed the retraction of liver apoptotic changes. They concluded that VK2 can be presumed to  be an effective nutritional supplement to aging population to spare the liver amongst other body organs against aging-induced deleterious injury. 

The topic of this study is interesting, however, it can be accepted after major revision.

I have itemized my major concerns in the following paragraphs.

Abstract

Line 41: eliminate “:” and insert “,”

Thank you for your comment. Change has been made as suggested

Introduction :

Improve the aim of the study explaining also the choice of the various markers of inflammation, fibrosis, even with bibliographical references .... avoiding doing it in the result section.

Thank you very much for your suggestion. A paragraph has been added discussing markers of oxidative stress, inflammation and fibrosis used (lines 64-79). The corresponding sentences has been removed from the results section.

Materials and methods

Line 119: What is “MMP”? Delete if it is a mistake; 

Thank you for your comment. MMP was a mistake and has been deleted

Line 120 delete “:” and check throughout the  manuscript 

Thank you. Change has been made as suggested

Line 121: Please add information regarding the homogenization buffer 

Information regarding homogenization has been added and highlighted as following: Hepatic tissue homogenates (10 % w/v) were prepared in chilled 1.15 % KCl at pH 7.4 using a hand-held homogenizer (Omni international, USA)

Line 123 : Add “real-time” before RT-PCR because In this paragraph , the authors describe the real-time RT-PCR.

Thank you. Change has been made as suggested

Line 158-159: add information regarding the secondary antibodies

Thank you. Source and Catalog number for secondary antibody used  has been added.

Results

Line 188: “figure1” in bold

Thank you. Change has been made as suggested 

Line 238-240: Delete this sentence “ Nrf-2/Keap-1/HO-1 axis is a conserved intracellular defense pathway to combat oxidative stress” from this paragraph and transfer it in the introduction section.

Thank you. Change has been made as suggested.

Line 240-246: This paragraph is difficult to understand as a graph (HO-1 protein level) is missing from figure 4. In addition, the authors must choose whether to use the percentage or the increment and decrement fold for the mRNA levels, otherwise it creates confusion. 

If the authors add the graph, they must also insert the ELISA assay description in the methods section.

Thank you for your comment. The figure was mispositioned and has been centered. Fold change was used for mRNA levels and ELISA assay has been added to methods section

Line 254-256 and Line 271-272: Delete and Transfer these sentences in the introduction. 

Thank you. Change has been made as suggested 

Figure 2: There is a problem in this figure. What are the images in the left side of this figure ??? in the figure legends the authors describe only the images in the right side (A1-C2). If it is a mistake, delete these images or alternatively describe these in the figure legend and in the paragraph. 

Thank you for your careful revision. Figure 2 has been revised and re-labelled and figure legend has been updated describing both right and left panels

 Figure4: Where is C? Miss the graph of ELISA as describe in the figure legend

 Thank you for you comment. The figure has been centered to show figure C.

Figure 5: This figure in not clear in particular for immunohistochemistry where the different images were not specify in the figure legend or in the relative paragraph of the results section (i.e. A1 and A2 of the I section are adult control ???...)

Thank you for your comment. The figures have been revised and re-labeled and figure legend have been updated.   

Figure 6: Miss “I” and “II” in the figure 

Thank you for your comment. The figures have been revised and re-labeled and figure legend have been updated.   

Discussion

The discussion section  is too long especially in the first part concerning the effects of aging. 

Thank you. Discussion has been revised and summarized.

Line: 296-364 . Reduce the discussion summarizing some parts

Thank you for your comment. Change has been made as suggested. 

Line 345 : “TUNEL” and not “TUNNEL”

Thank you. Corrected.

Line 417: “(13)” is not correct for reference style. 

Thank you for your revision. The correct reference has been added.

Reviewer 2 Report

The manuscript entitled “Vitamin K (MK-7) Intercepts Keap-1/Nrf-2/Ho-1 pathway and  Hinders Inflammatory/Apoptotic signalling and Liver Aging in Naturally Aging Rat” submitted to Antioxidants by Dr. El-Sherbiny and co-workers presents interesting findings on the protective effects of vitamin K on hepatic aging. The study was well designed and performed but some aspects should be corrected.

Line 9. faculty or Faculty?

Line 32. “In the current study, natural aging induced a significant liver deterioration with significant oxidative, inflammatory, apoptotic and fibrotic changes” What oxidative changes were observed?

Line 33. “VK2 admin-33 istration on the other hand, improved liver functions,…” liver functions is too much ambiguous.

Line 100. What is the rat number per experimental group? The number of determinations should be mentioned in each figure legend, too.

Line 101. What was VK2 dose chosen?

Line 111. “fourth group” or fourth part?

Line 156. Primary/secondary antibody dilutions should be mentioned to help to reproduce the experiments.

Line 172. “hepatic tissue” is too much ambiguous.

Line 253. “Effect of VK2 on hepatic inflammation in aged rats” In my opinion hepatic inflammation was not measured.

Line 254. “…proinflammatory mediators including COX-2, iNOS immunostaining and TNF-α protein levels were evaluated”. COX-2 and iNOS are not inflammatory mediators.

Line 302. “he” What means?

Line 350. “…increased fibrosis…” The manuscript did not present direct findings of fibrosis. Thus, this statement should be reviewed along the manuscript.

Author Response

First, we would like to thank the reviewer for dedicating the time and the effort to review our work.  The reviewer’s suggestions were very valuable and significant for the revision and improvement of our article quality and provided very important guidance to our research. The points that were raised by the reviewer are addressed below

The manuscript entitled “Vitamin K (MK-7) Intercepts Keap-1/Nrf-2/Ho-1 pathway and  Hinders Inflammatory/Apoptotic signalling and Liver Aging in Naturally Aging Rat” submitted to Antioxidants by Dr. El-Sherbiny and co-workers presents interesting findings on the protective effects of vitamin K on hepatic aging. The study was well designed and performed but some aspects should be corrected.

Line 9. faculty or Faculty?

Thank you, corrected to Faculty

Line 32. “In the current study, natural aging induced a significant liver deterioration with significant oxidative, inflammatory, apoptotic and fibrotic changes” What oxidative changes were observed?

Thank you for your comment. Changes in Keap-1 and Nrf-2 mRNA levels in addition to Ho-1 concertation has been observed. Nrf-2/ Keap-1/HO-1 axis is a known conserved intracellular defense pathway to combat oxidative stress. Reduced Nrf-2/Keap-1/ HO-1 axis reflects disrupted cellular redox homeostasis.

Bellezza I, Giambanco I, Minelli A, Donato R. Nrf2-Keap1 signaling in oxidative and reductive stress. Biochim Biophys Acta Mol Cell Res. 2018;1865(5):721-733. doi:10.1016/j.bbamcr.2018.02.010

However, based on your comment, we changed this sentence to  natural aging induced a significant liver deterioration with disrupted Keap-1/Nrf-2/HO-1 axis, increased COX-2, iNOS and TNF-α expression, apoptotic and fibrotic changes

Line 33. “VK2 admin-33 istration on the other hand, improved liver functions,…” liver functions is too much ambiguous.

Thank you for your comment. Based on your suggestion, this sentence has been changed to “improved biochemical indices of liver function (total protein, albumin, ALT and AST”

Line 100. What is the rat number per experimental group? The number of determinations should be mentioned in each figure legend, too.

Thank you for your comment. The number of animals have been indicated in Methods section and in figure legends

Line 101. What was VK2 dose chosen?

Thank you for your comment. The dose was 30 mg/kg, 5 days per week for 17 months

Line 111. “fourth group” or fourth part?

Thank you. Group has been changed to part

Line 156. Primary/secondary antibody dilutions should be mentioned to help to reproduce the experiments.

Thank you for your comment. Dilutions have been added

Line 172. “hepatic tissue” is too much ambiguous.

Thank you, changed to hepatic tissue sections

Line 253. “Effect of VK2 on hepatic inflammation in aged rats” In my opinion hepatic inflammation was not measured.

Thank you for your comment. Assessed proteins have been specified and the subtitle has been changed to: Effect of VK2 on hepatic COX-2, iNOS and TNF-α

Line 254. “…proinflammatory mediators including COX-2, iNOS immunostaining and TNF-α protein levels were evaluated”. COX-2 and iNOS are not inflammatory mediators.

Thank you for your valuable comment. The sentence has been rephrased to: For assessment of inflammation among different experimental groups, COX-2, iNOS and proinflammatory cytokine TNF-α immunostaining were evaluated

Line 302. “he” What means?

Thank you for your comment. We apologize for this typo error that has been deleted.

Line 350. “…increased fibrosis…” The manuscript did not present direct findings of fibrosis. Thus, this statement should be reviewed along the manuscript.

Thank you for your comment. Fibrosis has been changed to markers of fibrosis through the manuscript